# Cooperative Electromagnetic Data Annotation via Low-Rank Matrix Completion

**Wei Zhang** [1,2]**, Jian Yang** [3]**, Qiang Li** [1]**, Jingran Lin** [1]**, Huaizong Shao** [1] **and Guomin Sun** [1,*]

[1] School of Information and Communication Engineering, University of Electronic Science and Technology of China, Chengdu 611731, China
[2] Science and Technology on Electronic Information Control Laboratory, Chengdu 610036, China
[3] Northern Institute of Electronic Equipment of China, Beijing 100089, China
[*] Correspondence: gms_6220062@uestc.edu.cn

**Abstract:** Electromagnetic data annotation is one of the most important steps in many signal processing applications, e.g., radar signal deinterleaving and radar mode analysis. This work considers cooperative electromagnetic data annotation from multiple reconnaissance receivers/platforms. By exploiting the inherent correlation of the electromagnetic signal, as well as the correlation of the observations from multiple receivers, a low-rank matrix recovery formulation is proposed for the cooperative annotation problem. Specifically, considering the measured parameters of the same emitter should be roughly the same at different platforms, the cooperative annotation is modeled as a low-rank matrix recovery problem, which is solved iteratively either by the rank minimization method or the maximum-rank decomposition method. A comparison of the two methods, with the traditional annotation method on both the synthetic and real data, is given. Numerical experiments show that the proposed methods can effectively recover missing annotations and correct annotation errors.

**Keywords:** data annotation completion; radar reconnaissance data; low-rank matrix recovery

## 1. Introduction

As radar has been widely used in the battlefield, radar signal reconnaissance plays an important role in electronic warfare (EW). Typically, the first step of the radar reconnaissance system is to annotate the intercepted radar pulses with some key parameters, such as pulse width, carrier frequency, pulse repetition interval, direction of arrival (DOA), etc., which is also known as pulse description word (PDW). By analyzing the range and variation characteristics of these parameters, the working mode and behavior of the radar can be recognized. Therefore, accurate annotation is one of the key steps for radar countermeasure [1,2]. However, with the appearance of advanced multi-function radar systems, the electromagnetic environment has become increasingly complex, and the annotation is facing unprecedented challenges [3]. Firstly the electromagnetic spectrum is congested, and the pulse density of radar signals surges. At present, the pulse density in a typical environment may exceed millions or even tens of millions per second. Secondly, the advanced radar transmitter is programmable, networked, and intelligent, which leads to agile and overlapping parameters. The traditional fixed pulse pattern (such as fixed carrier frequency, repeated frequency, and unmodulated pulses) tends to be replaced with more complex time-varying patterns in modern radar systems. In addition, to improve the anti-reconnaissance and anti-jamming capabilities, more complex inter-pulse modulation patterns are adopted, which makes it hard to accurately annotate the parameters from the interception; the strong antagonism between the two sides of the non-cooperative game and the high real-time response induce incomplete and even wrong characteristic parameters of radar signals obtained by reconnaissance. Therefore, how to accurately and stably annotate the parameters of radar pluses is crucial for radar countermeasures.

Apart from radar countermeasures, data annotation is also commonly encountered in other fields, e.g., image and text data processing. At present, most annotations still rely on traditional manual methods. Manual annotation is often labor-intensive, tedious, and inefficient due to differences in personal experience and a lack of effective information. The heuristic rule-based annotation method and the pattern matching-based annotation method are also commonly used in the field of image and text data processing [4–7]. The annotation method based on the heuristic rule has low accuracy and generality, and cannot add semantic annotations to all the extracted data [7]. The pattern matching method utilizes the pre-established pattern matching relationship to annotate the data in a complementary manner [8], but in general, it is difficult to guarantee the correctness of the matching relationship. In view of the above shortcomings, it is difficult to adapt the traditional annotation methods to the reconnaissance electromagnetic data obtained under non-cooperative and strong confrontation conditions. Moreover, the reconnaissance data obtained by multiple heterogeneous platforms often have problems such as poor data quality, low annotation rate, and a serious lack of annotation information, which presents an obstacle to subsequent analyses and processing. How to realize the automatic annotation efficiently and accurately is particularly important for radar countermeasures.

In this work, we consider that radar reconnaissance data are intercepted by multiple reconnaissance platforms, but due to interference and noisy environments, each platform may have only partial, incomplete annotations of the radar pulses. Our goal is to use these partial annotations to cooperatively obtain an accurate and complete annotation. To this end, we exploit two key observations, namely, (1) radar reconnaissance data are often inherently correlated in the time-frequency domain; (2) interceptions from multiple platforms are highly correlated since they are from the same target. Upon the above two observations, we expect that the collected data from multiple platforms should exhibit a certain low-rank structure. The low-rank representation in matrix form is an important data representation, which has been widely used in various research areas such as robust principal component analysis [8,9] and matrix completion [10–13]. It also can be used for image restoration combined with sparse optimization [14–16]. Low-rank matrix recovery can be regarded as a generalization of compressed sensing, that is, how to recover the original matrix using the observation data under the low-rank condition [17–19]. Based on the theory of completion and recovery of the low-rank matrix, the redundancy existing in data can be exploited to fill in the missing elements or correct the erroneous annotations. While low-rank matrix completion has been widely used in other fields, e.g., image recovery [20–24] and matrix completion [25–33], to the best of our knowledge we are not aware of any work on electronic reconnaissance data annotation, especially in radar countermeasure applications. In this work, we first formulate the cooperative annotation problem as a low-rank matrix completion problem and then two efficient optimization algorithms are developed; one is based on convex relaxation and the other is non-convex max-rank decomposition. Simulations on synthetic data and real data are provided to demonstrate the efficacy of the proposed methods by comparing them with the conventional method.

The outline of this paper is given as follows. In Section 2, the problem formulation is presented. In Section 3, a rank-minimization algorithm for annotation completion is proposed. In Section 4, a maximum-rank-decomposition algorithm is proposed. In Section 5, numerical comparisons of the two proposed methods with some state-of-the-art algorithms are given. In the end, Section 6 concludes the paper.

## 2. Problem Formulation

Suppose that there are $n_1$ reconnaissance receivers/platforms and $n_2$ emitters/targets, e.g., radars, in the observation area within a certain time range. For each target, there are $n_3$ measured parameters, including time, location (such as longitude, altitude, and height), speed, frequency band, signal intensity, etc. An illustration of the measured parameters is given in Table 1, which records the annotation information of different platforms, where "$**$" represents the received value of measured parameters.

**Table 1.** An illustration of annotation information of electronic reconnaissance data.

| Platform Label | Target Label | Feature 1 | Feature 2 | Feature 3 | Feature 4 | Feature 5 | $\cdots$ | Feature $n_3$ |
|---|---|---|---|---|---|---|---|---|
| | 1 | ** | ** | ** | ** | ** | $\cdots$ | ** |
| 1 | $\vdots$ | $\vdots$ | $\vdots$ | $\vdots$ | $\vdots$ | $\vdots$ | $\vdots$ | $\vdots$ |
| | $n_2$ | ** | ** | ** | ** | ** | $\cdots$ | ** |
| | $\vdots$ | $\vdots$ | $\vdots$ | $\vdots$ | $\vdots$ | $\vdots$ | $\cdots$ | $\vdots$ |
| $\vdots$ | $\vdots$ | $\vdots$ | $\vdots$ | $\vdots$ | $\vdots$ | $\vdots$ | $\cdots$ | $\vdots$ |
| | $\vdots$ | $\vdots$ | $\vdots$ | $\vdots$ | $\vdots$ | $\vdots$ | $\cdots$ | $\vdots$ |
| | 1 | ** | ** | ** | ** | ** | $\cdots$ | ** |
| $n_1$ | $\vdots$ | $\vdots$ | $\vdots$ | $\vdots$ | $\vdots$ | $\vdots$ | $\vdots$ | $\vdots$ |
| | $n_2$ | ** | ** | ** | ** | ** | $\cdots$ | ** |

The characteristics of the targets observed at different platforms in Table 1 can be written as a matrix $X \in R^{m_1 \times n_3}$ by arranging measured parameters in the order of platforms, where $m_1 = n_1 \times n_2$.

$$X = \begin{bmatrix} x_{1,1} & x_{1,2} & \cdots & x_{1,n_3} \\ x_{2,1} & x_{2,2} & \cdots & x_{2,n_3} \\ \vdots & \vdots & \ddots & \vdots \\ x_{m_1,1} & x_{m_1,2} & \cdots & x_{m_1,n_3} \end{bmatrix} \tag{1}$$

In general, it is difficult to collect target information all the time at each platform, and the parameters (annotation information) detected by different platforms are not exactly the same due to the heterogeneous characteristics between different types of platforms. In addition, different platforms have different statuses, such as "work/maintenance", at the same time. All these facts lead to the missing characteristic information in Table 1 and matrix $X$, which is shown in Figure 1, where the small black squares represent the missing annotation information. Our goal is to recover the missing elements in the matrix $X$ from the partially observed data, i.e., annotation completion.

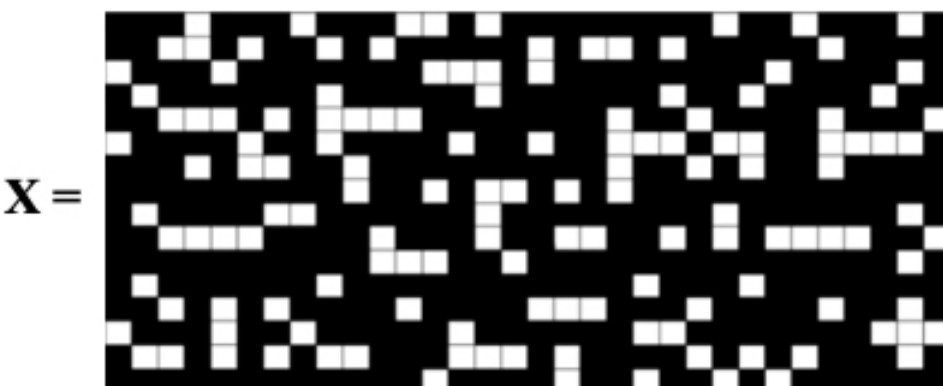

**Figure 1.** Partially annotated characteristics matrix.

According to the definition of $X$, the row vectors of characteristic parameters belonging to the same target should be highly correlated; therefore, the rank of matrix $X$ does not exceed the number of targets $n_2$, i.e., $r = \text{rank}(X) \le n_2$. The matrix $X$ is low-rank if there are enough monitoring platforms and enough categories of characteristic parameters, i.e., $r = \text{rank}(X) \ll \min\{m_1, n_3\}$. Thus, the annotation completion can be formulated as a low-rank matrix recovery problem, in which each row or column of the matrix can be expressed linearly by other rows or columns. The missing data can be recovered perfectly with a high

probability [10,22,23] using the redundant information when the rank of the matrix and the number of known elements meet certain conditions. Therefore, it is theoretically feasible to use the low-rank matrix recovery theory for annotation completion. To put it into context, let $D \in R^{m_1 \times n_3}$ be the observation matrix of $X$, which contains the known annotation information of $X$. The annotation completion problem based on low-rank matrix recovery can be modeled as:

$$\min_{X \in R^{m_1 \times n_3}} \|X - D\|_0 \quad s.t. \quad \text{rank}(X) \leq n_3 \tag{2}$$

where $\|X - D\|_0$ is the $\ell_0$-norm of $X - D$, i.e., the number of non-zero elements in $X - D$. This is a complex non-convex optimization problem since the non-convex function $\| \cdot \|_0$ and the non-convex constraint on $\text{rank}(X)$. It is difficult to obtain the global optimal solution. In order to solve this problem, the min-rank-based convex approximation algorithm and the max-rank-decomposition-based non-convex algorithm are employed to find approximate solutions for problem (2).

We summarize the frequently used notations in Table 2.

**Table 2.** The notation of symbols.

| Notation | Explanation |
|---|---|
| $X$, $D$, $E$, $\Lambda$, $U$, $V$ | Matrix |
| $e$ $e$ | Vector |
| $x_{i,j}$, $n_1$, $n_2$, $n_3$, $c$ | Scalar |

## 3. The Rank-Minimization-Based Convex Approximation Algorithm

In this section, a rank-minimization-based convex algorithm is proposed to solve problem (2). First, let $\Omega \subseteq \{1, 2, \ldots, m_1\} \times \{1, 2, \ldots, n_3\}$ denote the set of indices associated with the known annotations in $X$. Define the linear projection operator $P_\Omega : R^{m_1 \times n_3} \to R^{m_1 \times n_3}$ as follows:

$$P_\Omega = \begin{cases} D_{i,j}, & (i, j) \in \Omega \\ 0, & (i, j) \notin \Omega \end{cases} \tag{3}$$

where $D_{i,j}$ represents the element in the $i$-th row and $j$-th column of matrix $D \in R^{m_1 \times n_3}$. Then, problem (2) can be recast as the following matrix rank minimization problem.

$$\min_{X \in R^{m_1 \times n_3}} \text{rank}(X) \quad s.t. \quad P_\Omega(X) = P_\Omega(D) \tag{4}$$

where $\text{rank}(\cdot)$ is the rank function. Problem (4) is still a non-convex problem. Here, we consider its convex relaxation. In fact, $\text{rank}(X)$ describes the number of non-zero singular values of $X$, i.e., the $\ell_0$-norm of the singular value vector. Since the $\ell_0$-norm is a non-convex function, the $\ell_1$-norm is utilized as the convex approximation of $\ell_0$-norm, which gives rise to the nuclear norm of $X$ as the convex approximation of $\text{rank}(X)$. By introducing the matrix slack variable $E \in R^{m_1 \times n_3}$, the problem (4) can be approximated as the following convex problem

$$\min_{X, E \in R^{m_1 \times n_3}} \|X\|_* \quad s.t. \quad X + E = D, \quad P_\Omega(E) = 0 \tag{5}$$

where $\|X\|_*$ is the nuclear norm of $X$. To solve problem (5), we employ the alternating direction method of multiple (ADMM) algorithms. Specifically, denote the augmented Lagrangian function $L_c(X, E, \Lambda)$

$$L_c(X, E, \Lambda) = \|X\|_* + \text{Tr}\{\Lambda^T(D - X - E)\} + \frac{c}{2}\|D - X - E\|_F^2 \tag{6}$$

where $c > 0$ is the penalty factor, $\Lambda \in R^{m_1 \times n_3}$ is the Lagrangian multiplier matrix, $\text{Tr}\{\cdot\}$ is the trace of the matrix, $\| \cdot \|_F$ is the Frobenius norm. Then, problem (5) can be solved by alternately updating $X$, $E$, and $\Lambda$, respectively, as follows

$$
\begin{cases}
X_{k+1} = \arg \min_{X \in R^{m_1 \times n_3}} L_c(X, E_k, \Lambda_k) \\
E_{k+1} = \arg \min_{P_\Omega(E)=0} L_c(X_{k+1}, E, \Lambda_k) \\
\Lambda_{k+1} = \Lambda_k + c(D - X_{k+1} - E_{k+1}).
\end{cases}
\tag{7}
$$

In the following, the updating for (7) is given.

### 3.1. Updating X

The updating of $X \in R^{m_1 \times n_3}$ is conducted by solving the following problem (8).

$$
\min_X \|X\|_* - \text{Tr}\{\Lambda_k^T X\} + \frac{c}{2}\|X + E_k - D\|_F^2.
\tag{8}
$$

In order to solve (8), an auxiliary variable matrix $A_k \in R^{m_1 \times n_3}$ is introduced, which is defined as

$$
A_k = D - E_k + \frac{1}{c}\Lambda_k
\tag{9}
$$

and the singular value decomposition of $A_k$ is given by

$$
A_k = U_k \Sigma_k V_k^T
\tag{10}
$$

where $U_k \in R^{m_1 \times m_1}$ and $V_k \in R^{n_3 \times n_3}$ are the left and right singular matrices, respectively, $\Sigma_k \in R^{m_1 \times n_3}$ and $\Sigma_k = \text{Diag}\{\sigma_i\}, i = 1, 2, \ldots, \min\{m_1, n_3\}$ is a diagonal matrix with the diagonal elements $\sigma_i$ being the $i$-th singular value of $A_k$. Define the operator $[\cdot]^+$ as

$$
[\cdot]^+ = \max\{\cdot, 0\}.
\tag{11}
$$

Then, the optimal solution of problem (8) is given by [28]

$$
X_{k+1} = U_k \text{Diag}\left\{ [\sigma_i - c^{-1}]^+ \right\} V_k^T, \quad i = 1, 2, \ldots, \min\{m_1, n_3\}
\tag{12}
$$

### 3.2. Updating E

The updating of $E \in R^{m_1 \times n_3}$ can be given by solving

$$
\begin{aligned}
\min_{E \in R^{m_1 \times n_3}} \quad & \|E - (D - X_{k+1} + c^{-1}\Lambda_k)\|_F^2 \\
s.t. \quad & P_\Omega(E) = 0.
\end{aligned}
\tag{13}
$$

Clearly, the optimal solution $E_{k+1}$ of problem (13) is given by $D - X_{k+1} + c^{-1}\Lambda_k$ for elements not in the set $\Omega$, thus we have

$$
E_{k+1} = P_{\Omega'}(D - X_{k+1} + c^{-1}\Lambda_k)
\tag{14}
$$

where

$$
P_{\Omega'}(A_{i,j}) = \begin{cases} A_{i,j}, & (i,j) \notin \Omega \\ 0, & (i,j) \in \Omega \end{cases}
$$

Then, the whole procedure for solving problem (5) is summarized in Algorithm 1.

---

**Algorithm 1** The rank-minimization-based algorithm

---

**Initialization:** $D$, $X_0$, $E_0$, $\Lambda_0$, $k = 0$

**Repeat**

$\quad X_{k+1} = U_k \text{Diag}\{[\sigma_k - c^{-1}]^+\} V_k^T;$

$\quad E_{k+1} = P_\Omega (D - X_{k+1} + c^{-1}\Lambda_k);$

$\quad \Lambda_{k+1} = \Lambda_k + c(D - X_{k+1} - E_{k+1});$

$\quad k = k + 1;$

**Until** some stopping criteria satisfied;

**Return** $X_k$.

---

From Algorithm 1, we find that the computation consumption is mainly in updating matrix $X$ due to the singular value decomposition of $A_k$. The total computation complexity of Algorithm 1 is at the order of $O(\max\{m_1, n_3\}^3)$ since the size of $A_k$ is $m_1 \times n_3$.

## 4. The Maximum-Rank-Decomposition-Based Non-Convex Algorithm

In this section, we consider an alternative way to tackle the annotation completion problem (2) from the maximum-rank decomposition perspective. Specifically, the maximum-rank decomposition of $X \in R^{m_1 \times n_3}$ (suppose rank$(X) = m_2$) is given by

$$X = UV \tag{15}$$

where $U \in R^{m_1 \times m_2}$, $V \in R^{m_2 \times n_3}$. Upon (15), problem (2) is recast as

$$\min_{X \in R^{m_1 \times n_3}, U \in R^{m_1 \times m_2}, V \in R^{m_2 \times n_3}} \|X - D\|_1 \quad s.t. \quad X = UV \tag{16}$$

As before, we employ the ADMM approach to handle problem (15). Specifically, the augmented Lagrangian function of (16) is given as

$$L_c(X, U, V, \Phi) = \|X - D\|_1 + \text{Tr}\{\Phi^T(UV - X)\} + \frac{c}{2}\|UV - X\|_F^2 \tag{17}$$

where $\Phi \in R^{m_1 \times n_3}$ is the Lagrangian multiplier matrix, $c$ is the penalty factor. The ADMM algorithm repeatedly runs the following updating

$$\begin{cases} X_{k+1} = \arg \min_{X \in R^{m_1 \times n_3}} L_c(X, U_k, V_k, \Phi_k) \\ U_{k+1} = \arg \min_{U \in R_{m_1 \times m_2}} L_c(X_{k+1}, U, V_k, \Phi_k) \\ V_{k+1} = \arg \min_{V \in R^{m_2 \times n_3}} L_c(X_{k+1}, U_{k+1}, V, \Phi_k) \\ \Phi_{k+1} = \Phi_k + c(U_{k+1}V_{k+1} - X_{k+1}) \end{cases} \tag{18}$$

until stopping criteria are satisfied.

### 4.1. Updating X

The updating of $X$ is given by solving

$$\min_{X \in R^{m_1 \times n_3}} \|X - D\|_1 - \text{Tr}\{\Phi_k^T X\} + \frac{c}{2}\|U_k V_k - X\|_F^2. \tag{19}$$

By using the first-order optimality condition, we have

$$\Phi_k + c(U_k V_k - X) \in \partial \|X - D\|_1 \tag{20}$$

where $\partial \|X - D\|_1$ represents the sub-differential of $\|X - D\|_1$, which is given by

$$\partial \|X - D\|_1 = \begin{cases} \frac{X-D}{\|X-D\|_1}, & X \neq D \\ \{ee \mid \|ee\|_1 \leq 1\}, & X = D \end{cases} \tag{21}$$

with $ee \in R^{m_1 \times 1}$ and $\|ee\|_1 \leq 1$. Then, we have

$$X_{k+1} = \begin{cases} D, & \|Y_k\|_1 \leq 1 \\ \frac{\|Y_k\|_1 - 1}{c} \cdot \frac{Y_k}{\|Y_k\|_1 + D}, & otherwise \end{cases} \tag{22}$$

where $Y_k = \Phi_k + c(U_k V_k - D)$.

*4.2. Updating U*

The updating of $U \in R^{m_1 \times m_2}$ is given by solving

$$\min_{U \in R^{m_1 \times m_2}} \operatorname{Tr}\{\Phi_k^T U V_k\} + \frac{c}{2} \|U V_k - X_{k+1}\|_F^2. \tag{23}$$

As the problem (23) is an unconstrained quadratic program, the optimal solution can be given by the first-order optimality condition, thus we have

$$U_{k+1} = (X_{k+1} - \frac{1}{c}\Phi_k) V_k^T (V_k V_k^T)^{-1}. \tag{24}$$

*4.3. Updating V*

The $V \in R^{m_2 \times n_3}$ updating is given by solving

$$\min_{V \in R^{m_2 \times n_3}} \operatorname{Tr}\{\Phi_k^T U_{k+1} V\} + \frac{c}{2} \|U_{k+1} V - X_{k+1}\|_F^2. \tag{25}$$

Similar to the problem (23), its optimal solution is given by

$$V_{k+1} = (U_{k+1}^T U_{k+1})^{-1} U_{k+1}^T (X_{k+1} - \frac{1}{c}\Phi_k). \tag{26}$$

We summarize the whole procedure of the ADMM algorithm for problem (16) in Algorithm 2.

The computation complexity of Algorithm 2 is decided by the updating steps. Note that the size of $U_k$ is ($m_1 \times m_2$), the size of $V_k$ is ($m_2 \times n_3$), and according to the low-rank assumption, we have $m_2 \ll m_1$ and $m_2 \ll n_3$. The computation complexity for updating $X_k$, $U_k$, and $V_k$ is at the order of $O(m_1 \times m_2 \times n_3)$. It can be seen that the non-convex algorithm (Algorithm 2) has lower per-iteration complexity as compared with the convex algorithm (Algorithm 1).

In addition, two proposed methods are designed to recover the missing feature parameters, the value of parameters is real and the auxiliary variables using the algorithm are real as well. Therefore, they cannot be utilized for complex parameters directly.

---

**Algorithm 2** The max-rank-decomposition-based algorithm

---

**Initialization:** $D$, $U_0$, $V_0$, $\Phi_0$, k=0

**Repeat:**

$$X_{k+1} = \begin{cases} D, & \|Y_k\|_1 \leq 1 \\ \frac{\|Y_k\|_1 - 1}{c} \cdot \frac{Y_k}{\|Y_k\|_1 + D}, & otherwise \end{cases};$$

$$U_{k+1} = (X_{k+1} - \tfrac{1}{c}\Phi_k)V_k^T(V_kV_k^T)^{-1};$$

$$V_{k+1} = (U_{k+1}^T U_{k+1})^{-1}U_{k+1}^T(X_{k+1} - \tfrac{1}{c}\Phi_k);$$

$$k = k+1;$$

**Until** some stopping criteria satisfied;

**Return:** $X_k$.

---

## 5. Numerical Experiments and Discussion

In this section, the performance of the two proposed methods is tested with synthetic data and real data, and the comparison testing with three different methods is also given. To evaluate the performance, the mean squared error (MSE) is adopted as performance metrics, which is denoted as

$$\text{MSE} = \sqrt{\frac{\text{Error}}{m * n}}$$

with

$$\text{Error} = \sum_{i,j} \frac{\|X_{i,j} - \hat{X}_{i,j}\|^2}{\|X_{i,j}\|^2}$$

where $X$ is the original matrix with size $(m \times n)$, and $i = 1, 2, \ldots, m$, $j = 1, 2, \ldots, n$, $\hat{X}$ is the recovered matrix.

### 5.1. Synthetic Data Test of Proposed Methods

The synthetic data is generated by a radar target simulator, including 10 platforms, 10 targets in $t = (t_1, \ldots, t_{10})$, for each target, 10 features are utilized, and each feature is normalized, which forms the original data matrix $X$ with [100 × 100] and rank $r = 10$. In order to test the performance of proposed methods under different missing ratios, the observation matrix $D$ is given by randomly dropping out elements with different ratios in each row of $X$ and setting them as empty. Part of the elements of $X$ are shown in Table 3 and part of the observation matrix $D$ with 50% of the annotations of $X$ randomly removed is shown in Table 4.

In Tables 5 and 6, the completed annotations by Algorithms 1 and 2 are given respectively. It can be seen that the missing elements are recovered after matrix completion. Compared with the original matrix $X$, we found that the proposed methods can recover $X$ efficiently. Take the first row of X for example, the fourth, fifth, and sixth elements in Table 5 are recovered by Algorithm 1 with values 1.1639, 1.2384, and 1.0438, which are exactly the same as that in $X$; i.e., they are perfectly recovered. Meanwhile, the corresponding recovered values by Algorithm 2 in Table 6 are 1.1643, 1.1978, and 1.0437, with MSE $\leq 1 \times 10^{-3}$, which suggests that the proposed methods can fill in the missing annotations efficiently.

**Table 3.** The original annotated matrix *X*.

| Target Label ($t_i$) | Feature 1 | Feature 2 | Feature 3 | Feature 4 | Feature 5 | Feature 6 | Feature 7 | Feature 8 | Feature 9 | Feature 10 | $\cdots$ |
|---|---|---|---|---|---|---|---|---|---|---|---|
| 1 | 0.8331 | 0.9314 | 1.6636 | 1.1639 | 1.2384 | 1.0438 | 1.2527 | 1.0609 | 0.5221 | 0.8351 | $\cdots$ |
| 2 | 0.7860 | 1.3702 | 1.6861 | 1.6636 | 1.2148 | 0.8691 | 1.1024 | 1.7871 | 0.7318 | 1.1431 | $\cdots$ |
| 3 | 1.0400 | 0.9844 | 1.1685 | 1.1966 | 0.9242 | 0.6846 | 1.0263 | 1.0460 | 0.6473 | 0.8802 | $\cdots$ |
| 4 | 0.7558 | 1.1816 | 1.4044 | 1.5881 | 1.0996 | 0.7906 | 0.9751 | 1.7054 | 0.7131 | 0.9999 | $\cdots$ |
| 5 | 1.1372 | 1.5230 | 2.2505 | 2.0789 | 1.7841 | 1.4639 | 1.6405 | 2.1031 | 0.9132 | 1.3650 | $\cdots$ |
| 6 | 0.6587 | 0.8033 | 1.5688 | 1.2180 | 1.2643 | 1.1109 | 1.1562 | 1.2050 | 0.4966 | 0.7463 | $\cdots$ |
| 7 | 0.2884 | 0.5634 | 0.5258 | 0.6923 | 0.4723 | 0.4630 | 0.3422 | 0.7309 | 0.2969 | 0.5746 | $\cdots$ |
| 8 | 0.7313 | 0.8509 | 0.9388 | 1.1082 | 0.7031 | 0.3967 | 0.7450 | 1.1216 | 0.5658 | 0.6677 | $\cdots$ |
| 9 | 1.0431 | 1.3981 | 1.7029 | 1.7407 | 1.3799 | 1.1854 | 1.2670 | 1.7087 | 0.8046 | 1.3185 | $\cdots$ |
| 10 | 1.0597 | 1.3580 | 1.6473 | 1.9323 | 1.3356 | 0.9283 | 1.2509 | 2.0165 | 0.9090 | 1.1356 | $\cdots$ |
| 1 | 1.7201 | 1.7509 | 2.6238 | 2.1080 | 2.0970 | 1.9122 | 2.1279 | 1.7904 | 1.0234 | 1.7243 | $\cdots$ |
| 2 | 0.7612 | 1.3054 | 2.0316 | 1.5709 | 1.5117 | 1.3303 | 1.13453 | 1.6025 | 0.6299 | 1.2025 | $\cdots$ |
| 3 | 0.7386 | 0.9366 | 1.3114 | 1.1991 | 1.1835 | 1.2654 | 1.0069 | 1.0870 | 0.5154 | 1.0572 | $\cdots$ |
| 4 | 0.7747 | 1.0134 | 1.7442 | 1.3499 | 1.4382 | 1.4058 | 1.2767 | 1.2729 | 0.5502 | 1.0479 | $\cdots$ |
| 5 | 1.1288 | 1.1637 | 1.5743 | 1.2103 | 1.1755 | 1.0365 | 1.2772 | 0.9698 | 0.6280 | 1.1144 | $\cdots$ |
| $\vdots$ | $\vdots$ | $\vdots$ | $\vdots$ | $\vdots$ | $\vdots$ | $\vdots$ | $\vdots$ | $\vdots$ | $\vdots$ | $\vdots$ | $\ddots$ |

**Table 4.** The partially annotated matrix *D*.

| Target Label ($t_i$) | Feature 1 | Feature 2 | Feature 3 | Feature 4 | Feature 5 | Feature 6 | Feature 7 | Feature 8 | Feature 9 | Feature 10 | $\cdots$ |
|---|---|---|---|---|---|---|---|---|---|---|---|
| 1 | 0.8331 |  | 1.6636 |  |  |  | 1.2527 |  | 0.5221 | 0.8351 | $\cdots$ |
| 2 | 0.7860 | 1.3702 |  | 1.6636 | 1.2148 |  |  | 1.7871 | 0.7318 | 1.1431 | $\cdots$ |
| 3 |  | 0.9844 | 1.1685 |  | 0.9242 |  | 1.0263 |  |  |  | $\cdots$ |
| 4 | 0.7558 |  | 1.4044 |  |  |  |  | 1.7054 | 0.7131 | 0.9999 | $\cdots$ |
| 5 | 1.1372 |  |  |  | 1.7841 | 1.4639 |  | 2.1031 | 0.9132 | 1.3650 | $\cdots$ |
| 6 |  |  | 1.5688 |  | 1.2643 |  |  | 1.2050 | 0.4966 | 0.7463 | $\cdots$ |
| 7 | 0.2884 |  | 0.5258 | 0.6923 | 0.4723 | 0.4630 | 0.3422 | 0.7309 | 0.2969 | 0.5746 | $\cdots$ |
| 8 | 0.7313 | 0.8509 |  |  |  |  | 0.7450 |  | 0.5658 | 0.6677 | $\cdots$ |
| 9 | 1.0431 |  |  |  | 1.3799 |  | 1.2670 |  |  | 1.3185 | $\cdots$ |
| 10 |  |  | 1.6473 | 1.9323 |  |  | 1.2509 | 2.0165 | 0.9090 |  | $\cdots$ |
| 1 |  |  | 2.6238 |  | 2.0970 | 1.9122 | 2.1279 |  |  | 1.7243 | $\cdots$ |
| 2 | 0.7612 | 1.3054 |  |  | 1.5117 |  |  |  | 0.6299 | 1.2025 | $\cdots$ |
| 3 |  |  | 1.3114 | 1.1991 |  |  | 1.0069 |  | 0.5154 | 1.0572 | $\cdots$ |
| 4 |  | 1.0134 |  |  | 1.4382 | 1.4058 |  | 1.2729 | 0.5502 |  | $\cdots$ |
| 5 |  | 1.1637 |  | 1.2103 | 1.1755 | 1.0365 | 1.2772 | 0.9698 | 0.6280 | 1.1144 | $\cdots$ |
| $\vdots$ | $\vdots$ | $\vdots$ | $\vdots$ | $\vdots$ | $\vdots$ | $\vdots$ | $\vdots$ | $\vdots$ | $\vdots$ | $\vdots$ | $\ddots$ |

In Figure 2, the MSE of two proposed methods under different missing rates is given. It can be found that the MSE decreases with the decreasing of the missing ratio, which suggests that both of the proposed methods can recover or recorrect the missing or wrong elements in *D* efficiently. Comparing the two methods, we find that Algorithm 1 has lower MSE with the missing ratio < 0.7, the main reason is that the completion by max rank decomposition in Algorithm 2 results in the measurement error.

**Table 5.** Results recovered by Algorithm 1.

| Target Label ($t_i$) | Feature 1 | Feature 2 | Feature 3 | Feature 4 | Feature 5 | Feature 6 | Feature 7 | Feature 8 | Feature 9 | Feature 10 | $\cdots$ |
|---|---|---|---|---|---|---|---|---|---|---|---|
| 1 | 0.8331 | 0.9314 | 1.6636 | 1.1639 | 1.2384 | 1.0438 | 1.2527 | 1.0609 | 0.5221 | 0.8351 | $\cdots$ |
| 2 | 0.7860 | 1.3702 | 1.6861 | 1.6636 | 1.2148 | 1.1024 | 1.7871 | 1.7871 | 0.7318 | 1.1431 | $\cdots$ |
| 3 | 1.0400 | 0.9844 | 1.1685 | 1.1966 | 0.9242 | 0.6946 | 1.0263 | 1.0460 | 0.6473 | 0.8802 | $\cdots$ |
| 4 | 0.7558 | 1.1816 | 1.4044 | 1.5881 | 1.0996 | 0.7906 | 0.9751 | 1.7054 | 0.7131 | 0.9999 | $\cdots$ |
| 5 | 1.1372 | 1.5230 | 2.2505 | 2.0789 | 1.7841 | 1.4639 | 1.6405 | 2.1031 | 0.9132 | 1.3650 | $\cdots$ |
| 6 | 0.6587 | 0.8033 | 1.5688 | 1.2180 | 1.2643 | 1.1109 | 1.1562 | 1.2050 | 0.4966 | 0.7463 | $\cdots$ |
| 7 | 0.2884 | 0.5634 | 0.5258 | 0.6923 | 0.4723 | 0.4630 | 0.3422 | 0.7309 | 0.2969 | 0.5746 | $\cdots$ |
| 8 | 0.7313 | 0.8509 | 0.9388 | 1.1082 | 0.7031 | 0.3967 | 0.7450 | 1.1216 | 0.5658 | 0.6677 | $\cdots$ |
| 9 | 1.0431 | 1.3981 | 1.7029 | 1.7407 | 1.3799 | 1.1854 | 1.2670 | 1.7087 | 0.8046 | 1.3185 | $\cdots$ |
| 10 | 1.0597 | 1.3580 | 1.6473 | 1.9323 | 1.3356 | 0.9283 | 1.2509 | 2.0165 | 0.9090 | 1.1356 | $\cdots$ |
| 1 | 1.7201 | 1.7509 | 2.6238 | 2.1080 | 2.0970 | 1.9122 | 2.1279 | 1.7904 | 1.0234 | 1.7243 | $\cdots$ |
| 2 | 0.7612 | 1.3054 | 2.0316 | 1.5709 | 1.5117 | 1.3303 | 1.3453 | 1.6025 | 0.6299 | 1.2025 | $\cdots$ |
| 3 | 0.7386 | 0.9366 | 1.3114 | 1.1991 | 1.1835 | 1.2654 | 1.0069 | 1.0870 | 0.5154 | 1.0572 | $\cdots$ |
| 4 | 0.7747 | 1.0134 | 1.7442 | 1.3499 | 1.4382 | 1.4058 | 1.2767 | 1.2729 | 0.5502 | 1.0479 | $\cdots$ |
| 5 | 1.1288 | 1.1637 | 1.5743 | 1.2103 | 1.1755 | 1.0365 | 1.2772 | 0.9698 | 0.6280 | 1.1144 | $\cdots$ |
| $\vdots$ | $\vdots$ | $\vdots$ | $\vdots$ | $\vdots$ | $\vdots$ | $\vdots$ | $\vdots$ | $\vdots$ | $\vdots$ | $\vdots$ | $\ddots$ |

**Table 6.** Results recovered by Algorithm 2.

| Target Label ($t_i$) | Feature 1 | Feature 2 | Feature 3 | Feature 4 | Feature 5 | Feature 6 | Feature 7 | Feature 8 | Feature 9 | Feature 10 | $\cdots$ |
|---|---|---|---|---|---|---|---|---|---|---|---|
| 1 | 0.8331 | 0.9376 | 1.6636 | 1.1643 | 1.1978 | 1.0437 | 1.2527 | 1.0901 | 0.5221 | 0.8351 | $\cdots$ |
| 2 | 0.7860 | 1.3702 | 1.6875 | 1.6636 | 1.2148 | 0.8691 | 1.1693 | 1.7420 | 0.7318 | 1.1431 | $\cdots$ |
| 3 | 1.0413 | 0.9844 | 1.1685 | 1.1685 | 0.9242 | 0.6946 | 1.0263 | 1.1058 | 0.6496 | 0.8788 | $\cdots$ |
| 4 | 0.7558 | 1.0976 | 1.4044 | 1.5837 | 1.1597 | 0.7942 | 1.1111 | 1.7054 | 0.7131 | 0.9999 | $\cdots$ |
| 5 | 1.1372 | 1.6160 | 2.2437 | 2.0659 | 1.7841 | 1.4639 | 1.6631 | 2.1031 | 0.9132 | 1.3650 | $\cdots$ |
| 6 | 0.6683 | 0.8156 | 1.5688 | 1.2899 | 1.2643 | 1.1103 | 1.0857 | 1.2050 | 0.4966 | 0.7463 | $\cdots$ |
| 7 | 0.2884 | 0.5617 | 0.5258 | 0.6923 | 0.4723 | 0.4630 | 0.3422 | 0.7309 | 0.2969 | 0.5746 | $\cdots$ |
| 8 | 0.7313 | 0.8509 | 0.9844 | 1.1183 | 0.7166 | 0.6775 | 0.7450 | 1.1288 | 0.5658 | 0.6677 | $\cdots$ |
| 9 | 1.0431 | 1.3351 | 1.7504 | 1.6924 | 1.3799 | 0.4025 | 1.2670 | 1.5447 | 0.7677 | 1.3185 | $\cdots$ |
| 10 | 1.0970 | 1.3589 | 1.6473 | 1.9323 | 1.3597 | 0.2262 | 1.2509 | 2.0165 | 0.9090 | 1.2495 | $\cdots$ |
| 1 | 1.7189 | 1.8459 | 2.6238 | 2.1156 | 2.0970 | 1.9122 | 2.1279 | 1.7915 | 1.1190 | 1.7243 | $\cdots$ |
| 2 | 0.7612 | 1.3054 | 1.9913 | 1.5768 | 1.5117 | 1.2374 | 1.4113 | 1.5867 | 0.6299 | 1.2025 | $\cdots$ |
| 3 | 0.8057 | 0.9980 | 1.3114 | 1.1991 | 1.1721 | 1.2709 | 1.0069 | 1.1548 | 0.5154 | 1.0572 | $\cdots$ |
| 4 | 0.7834 | 1.0134 | 1.5256 | 1.4227 | 1.4382 | 1.4058 | 1.1975 | 1.2729 | 0.5502 | 1.0699 | $\cdots$ |
| 5 | 0.9414 | 1.1637 | 1.5289 | 1.2103 | 1.1755 | 1.0365 | 1.2772 | 0.9698 | 0.6280 | 1.1144 | $\cdots$ |
| $\vdots$ | $\vdots$ | $\vdots$ | $\vdots$ | $\vdots$ | $\vdots$ | $\vdots$ | $\vdots$ | $\vdots$ | $\vdots$ | $\vdots$ | $\ddots$ |

In the discussion above, we have assumed $\text{rank}(X) = 10$ as a prior. In practice, the rank of $X$ is generally unknown and needs to be jointly estimated. In fact, the rank minimization in Algorithm 1 cannot estimate the rank of $D$ directly, while Algorithm 2 can predict the rank directly due to the max-rank decomposition of $D$. The comparison of the estimated rank and the real rank of $X$ given by Algorithm 2 is presented in Figure 3. It can be seen that the estimated rank of the proposed method is consistent with the real rank. In fact, we find that when the missing ratio $\leq 50\%$, the curve of rank setting vs. estimated rank is consistent with the curve in Figure 3. The main reason is that fewer missing records result in better recovery results. When the missing ratio is $\geq 50\%$, the estimated rank is unstable and not consistent with the rank setting, the main reason is that more missing records can lead to rank variation.

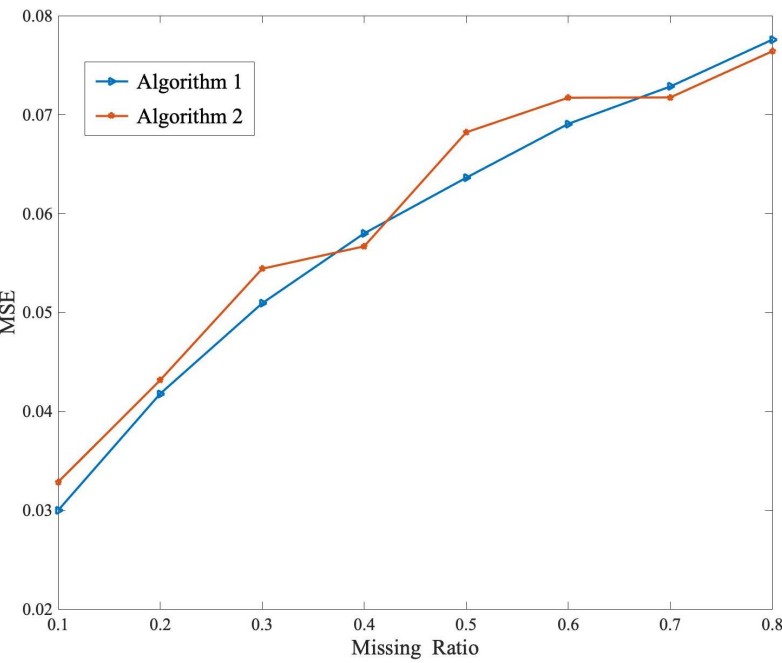

**Figure 2.** The MSE of two proposed methods under different missing ratios.

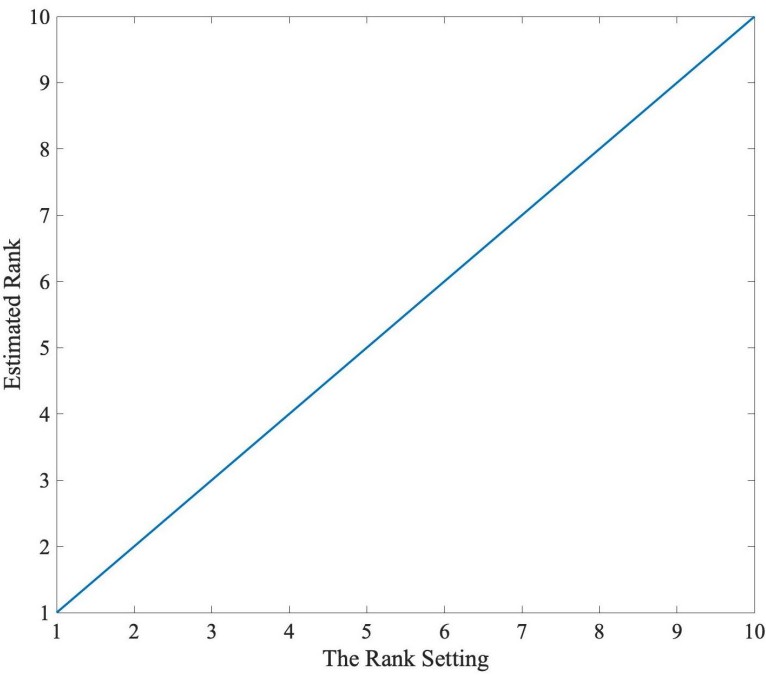

**Figure 3.** The rank setting vs. estimated rank of Algorithm 2.

## 5.2. Real Data Test of Proposed Methods

Apart from the synthetic data test, in the following, we verify the performance of the proposed methods with real data—PDW records from real radars. For the real data test, the missing ratio is about 30%. The missing information is set as empty, moreover, certain errors are added to verify the error correction capability of proposed methods. Part of the real data $X$ and the observation data $D$ are illustrated in Tables 7 and 8, respectively.

**Table 7.** Real data $X$.

| Target | Platf-1 | | | | | Platf-2 | | | | | $\cdots$ |
| Label | FW | PW | PT | AM | AOA | FW | PW | PT | AM | AOA | $\cdots$ |
|---|---|---|---|---|---|---|---|---|---|---|---|
| 1 | $4.7654 \times 10^3$ | 2.1200 | 780 | 95.2850 | 428.7933 | $4.7655 \times 10^3$ | 1.7600 | 780 | 95.7800 | 428.7967 | $\cdots$ |
| 2 | $4.7653 \times 10^3$ | 2.1400 | 800 | 95.5050 | 428.8067 | $4.7657 \times 10^3$ | 1.8000 | 770 | 95.5050 | 428.7933 | $\cdots$ |
| 3 | $4.7655 \times 10^3$ | 1.7200 | 820 | 95.5600 | 428.7933 | $4.7656 \times 10^3$ | 1.9000 | 770 | 95.5050 | 428.7767 | $\cdots$ |
| 4 | $4.7656 \times 10^3$ | 1.8000 | 830 | 95.5600 | 428.7867 | $4.7655 \times 10^3$ | 1.9200 | 760 | 96.0750 | 428.8000 | $\cdots$ |
| 5 | $4.7656 \times 10^3$ | 1.9200 | 820 | 95.5600 | 428.7767 | $4.7655 \times 10^3$ | 2.1200 | 780 | 95.5050 | 428.7633 | $\cdots$ |
| 6 | $4.7653 \times 10^3$ | 2.0800 | 830 | 95.6700 | 428.7700 | $4.7656 \times 10^3$ | 2.1200 | 750 | 95.6700 | 428.7867 | $\cdots$ |
| 7 | $4.7654 \times 10^3$ | 2.1200 | 840 | 95.6700 | 428.7900 | $4.7655 \times 10^3$ | 1.7400 | 770 | 95.3950 | 428.7933 | $\cdots$ |
| 8 | $4.7655 \times 10^3$ | 2.2000 | 840 | 95.6700 | 428.7833 | $4.7656 \times 10^3$ | 1.7600 | 780 | 95.6700 | 428.7967 | $\cdots$ |
| 9 | $4.7656 \times 10^3$ | 2.2000 | 850 | 95.6150 | 428.8000 | $4.7656 \times 10^3$ | 1.9000 | 790 | 95.4500 | 428.7733 | $\cdots$ |
| 10 | $4.7656 \times 10^3$ | 1.9000 | 840 | 95.6150 | 428.7700 | $4.7656 \times 10^3$ | 1.9000 | 800 | 95.6150 | 428.8000 | $\cdots$ |
| 11 | $4.7656 \times 10^3$ | 1.9200 | 870 | 95.5600 | 428.8000 | $4.7656 \times 10^3$ | 1.9000 | 780 | 95.5050 | 428.8000 | $\cdots$ |
| 12 | $4.7656 \times 10^3$ | 1.9000 | 870 | 95.5050 | 428.8000 | $4.7655 \times 10^3$ | 1.7200 | 800 | 95.5600 | 428.8267 | $\cdots$ |
| 13 | $4.7653 \times 10^3$ | 2.0600 | 880 | 95.5600 | 428.7700 | $4.7657 \times 10^3$ | 1.9000 | 800 | 95.6700 | 428.7733 | $\cdots$ |
| 14 | $4.7655 \times 10^3$ | 2.1200 | 880 | 95.5600 | 428.7833 | $4.7656 \times 10^3$ | 1.8800 | 810 | 95.5050 | 428.8000 | $\cdots$ |
| 15 | $4.7655 \times 10^3$ | 2.1400 | 890 | 95.5600 | 428.8000 | $4.7656 \times 10^3$ | 1.7400 | 810 | 95.6700 | 428.7433 | $\cdots$ |
| 16 | $4.7656 \times 10^3$ | 2.1600 | 890 | 95.6150 | 428.7900 | $4.7656 \times 10^3$ | 1.8000 | 820 | 95.5600 | 428.7933 | $\cdots$ |
| 17 | $4.7656 \times 10^3$ | 2.1600 | 900 | 95.5600 | 428.7933 | $4.7656 \times 10^3$ | 1.9200 | 820 | 95.4500 | 428.7767 | $\cdots$ |
| 18 | $4.7656 \times 10^3$ | 1.9200 | 890 | 95.6150 | 428.7767 | $4.7656 \times 10^3$ | 1.8800 | 830 | 95.5600 | 428.8000 | $\cdots$ |
| 19 | $4.7610 \times 10^3$ | **0.2200** | 730 | 95.3400 | **0.3533** | $4.7656 \times 10^3$ | 1.8800 | 830 | 95.5600 | 428.8000 | $\cdots$ |
| 20 | $4.7654 \times 10^3$ | 1.9000 | 900 | 95.5050 | 428.4467 | $4.7655 \times 10^3$ | 2.1400 | 830 | 95.6150 | 428.7533 | $\cdots$ |
| $\vdots$ | $\vdots$ | $\vdots$ | $\vdots$ | $\vdots$ | $\vdots$ | $\vdots$ | $\vdots$ | $\vdots$ | $\vdots$ | $\vdots$ | $\ddots$ |

**Table 8.** Recorded real data $D$ with missing annotations.

| Target | Platf-1 | | | | | Platf-2 | | | | | ··· |
| Label | FW | PW | PT | AM | AOA | FW | PW | PT | AM | DOA | ··· |
|---|---|---|---|---|---|---|---|---|---|---|---|
| 1 | $4.7654 \times 10^3$ | 2.1200 | | 95.2850 | 0 | | | | 95.7800 | 428.7967 | ··· |
| 2 | $4.7653 \times 10^3$ | | 800 | 95.5050 | | $4.7657 \times 10^3$ | 1.8000 | | 95.5050 | 428.7933 | ··· |
| 3 | $4.7655 \times 10^3$ | | | 95.5600 | 428.7933 | $4.7656 \times 10^3$ | 1.9000 | | 95.5050 | 428.7767 | ··· |
| 4 | $4.7656 \times 10^3$ | 1.8000 | 830 | | 428.7867 | $4.7655 \times 10^3$ | 1.9200 | 760 | 96.0750 | 428.8000 | ··· |
| 5 | $4.7656 \times 10^3$ | | | 95.5600 | | $4.7655 \times 10^3$ | 2.1200 | 780 | 95.5050 | 428.7633 | ··· |
| 6 | $4.7653 \times 10^3$ | 2.0800 | 830 | | 428.7700 | $4.7656 \times 10^3$ | | | 95.6700 | 428.7867 | ··· |
| 7 | $4.7654 \times 10^3$ | | | 95.6700 | | | 1.7400 | 770 | | | ··· |
| 8 | $4.7655 \times 10^3$ | 2.2000 | 840 | 95.6700 | 428.7833 | $4.7656 \times 10^3$ | 1.7600 | | 95.6700 | | ··· |
| 9 | | 2.2000 | | | 428.8000 | $4.7656 \times 10^3$ | 1.9000 | 790 | | 428.7733 | ··· |
| 10 | $4.7656 \times 10^3$ | 1.9000 | 840 | 95.6150 | | $4.7656 \times 10^3$ | 1.9000 | 800 | 95.6150 | 428.8000 | ··· |
| 11 | $4.7656 \times 10^3$ | 1.9200 | | | 428.8000 | $4.7656 \times 10^3$ | 1.9000 | | 95.5050 | 428.8000 | ··· |
| 12 | $4.7656 \times 10^3$ | | 870 | | 428.8000 | $4.7655 \times 10^3$ | | 800 | 95.5600 | 428.8267 | ··· |
| 13 | $4.7653 \times 10^3$ | | | 95.5600 | 428.7700 | $4.7657 \times 10^3$ | | | | 428.7733 | ··· |
| 14 | | 2.1200 | 880 | | 428.7833 | $4.7656 \times 10^3$ | 1.8800 | 810 | 95.5050 | 428.8000 | ··· |
| 15 | $4.7655 \times 10^3$ | 2.1400 | | 95.5600 | 428.8000 | $4.7656 \times 10^3$ | | 810 | 95.6700 | 428.7433 | ··· |
| 16 | $4.7656 \times 10^3$ | 2.1600 | 890 | 95.6150 | 428.7900 | $4.7656 \times 10^3$ | 1.8000 | | | 428.7933 | ··· |
| 17 | $4.7656 \times 10^3$ | 2.1600 | 900 | 95.5600 | 428.7933 | $4.7656 \times 10^3$ | | 820 | 95.4500 | 428.7767 | ··· |
| 18 | $4.7656 \times 10^3$ | 1.9200 | 890 | | 428.7767 | | 1.8800 | 830 | 95.5600 | 428.8000 | ··· |
| 19 | $4.7610 \times 10^3$ | | 730 | | | $4.7656 \times 10^3$ | | 830 | 95.5600 | 428.8000 | ··· |
| 20 | $4.7654 \times 10^3$ | 1.9000 | | 95.5050 | | $4.7655 \times 10^3$ | 2.1400 | 830 | 95.6150 | 428.7533 | ··· |
| ⋮ | ⋮ | ⋮ | ⋮ | ⋮ | ⋮ | ⋮ | ⋮ | ⋮ | ⋮ | ⋮ | ⋱ |

The recovery for missing PDW rerecords of Algorithms 1 and 2 are shown in Tables 9 and 10, respectively. From the two tables, it can be seen that both methods can fill in the missing annotations accurately. Specifically, for the carrier frequency annotation in the first column, the MSE is $\leq 1 \times 10^{-3}$; for the pulse width annotation in the second column, the MSE is about $1 \times 10^{-2}$; for the amplitude annotation in the fourth column, the error is about $1 \times 10^{-2}$; for the AOA parameter in the last column, the error is about $1 \times 10^{-3}$.

**Table 9.** Results recovered by Algorithm 1.

| Target | Platf-1 | | | | | Platf-2 | | | | | ··· |
| Label | FW | PW | PT | AM | AOA | FW | PW | PT | AM | DOA | ··· |
|---|---|---|---|---|---|---|---|---|---|---|---|
| 1 | $4.7654 \times 10^3$ | 2.1200 | 862.3198 | 95.2850 | 0 | $4.7347 \times 10^3$ | 1.7002 | 769.1010 | 95.7800 | 428.7967 | ··· |
| 2 | $4.7653 \times 10^3$ | 1.8810 | 800 | 95.5050 | 590.2473 | $4.7657 \times 10^3$ | 1.8000 | 763.4213 | 95.5050 | 428.7933 | ··· |
| 3 | $4.7655 \times 10^3$ | 1.7387 | 813.6904 | 95.5600 | 428.7933 | $4.7656 \times 10^3$ | 1.9000 | 778.7780 | 95.5050 | 428.7767 | ··· |
| 4 | $4.7656 \times 10^3$ | 1.8000 | 830 | 93.3096 | 428.7867 | $4.7655 \times 10^3$ | 1.9200 | 760 | 96.0750 | 428.8000 | ··· |
| 5 | $4.7656 \times 10^3$ | 2.0022 | 839.8333 | 95.5600 | 481.1485 | $4.7655 \times 10^3$ | 2.1200 | 780 | 95.5050 | 428.7633 | ··· |
| 6 | $4.7653 \times 10^3$ | 2.0800 | 830 | 93.5123 | 428.7700 | $4.7656 \times 10^3$ | 1.7048 | 777.1674 | 95.6700 | 428.7867 | ··· |
| 7 | $4.7654 \times 10^3$ | 2.0342 | 848.9644 | 95.6700 | 243.2014 | $4.7000 \times 10^3$ | 1.7400 | 770 | 94.1180 | 543.5496 | ··· |
| 8 | $4.7655 \times 10^3$ | 2.2000 | 840 | 95.6700 | 428.7833 | $4.7656 \times 10^3$ | 1.7600 | 754.3832 | 95.6700 | 487.8920 | ··· |
| 9 | $4.6963 \times 10^3$ | 2.2000 | 789.7436 | 93.3603 | 428.8000 | $4.7656 \times 10^3$ | 1.9000 | 790 | 93.2589 | 428.7733 | ··· |
| 10 | $4.7656 \times 10^3$ | 1.9000 | 840 | 95.6150 | 146.4476 | $4.7656 \times 10^3$ | 1.9000 | 800 | 95.6150 | 428.8000 | ··· |
| 11 | $4.7656 \times 10^3$ | 1.9200 | 811.1999 | 93.4410 | 428.8000 | $4.7656 \times 10^3$ | 1.9000 | 756.8295 | 95.5050 | 428.8000 | ··· |
| 12 | $4.7656 \times 10^3$ | 2.0224 | 870 | 94.6997 | 428.8000 | $4.7655 \times 10^3$ | 1.7649 | 800 | 95.5600 | 428.8267 | ··· |
| 13 | $4.7653 \times 10^3$ | 1.8753 | 837.7079 | 95.5600 | 428.7700 | $4.7657 \times 10^3$ | 1.7816 | 784.7113 | 94.5026 | 428.7733 | ··· |
| 14 | $4.7233 \times 10^3$ | 2.1200 | 880 | 94.4242 | 428.7833 | $4.7656 \times 10^3$ | 1.8800 | 810 | 95.5050 | 428.8000 | ··· |
| 15 | $4.7655 \times 10^3$ | 2.1400 | 847.0041 | 95.5600 | 428.8000 | $4.7656 \times 10^3$ | 1.6587 | 810 | 95.6700 | 428.7433 | ··· |
| 16 | $4.7656 \times 10^3$ | 2.1600 | 890 | 95.6150 | 428.7900 | $4.7656 \times 10^3$ | 1.8000 | 784.9361 | 94.9520 | 428.7933 | ··· |
| 17 | $4.7656 \times 10^3$ | 2.1600 | 900 | 95.5600 | 428.7933 | $4.7656 \times 10^3$ | 1.7786 | 820 | 95.4500 | 428.7767 | ··· |
| 18 | $4.7656 \times 10^3$ | 1.9200 | 890 | 94.8011 | 428.7767 | $4.7250 \times 10^3$ | 1.8800 | 830 | 95.5600 | 428.8000 | ··· |
| 19 | $4.7610 \times 10^3$ | 1.7863 | 730 | 95.2033 | 461.7466 | $4.7656 \times 10^3$ | 1.8032 | 830 | 95.5600 | 428.8000 | ··· |
| 20 | $4.7654 \times 10^3$ | 1.9000 | 813.5963 | 95.5050 | 423.7582 | $4.7655 \times 10^3$ | 2.1400 | 830 | 95.6150 | 428.7533 | ··· |
| ⋮ | ⋮ | ⋮ | ⋮ | ⋮ | ⋮ | ⋮ | ⋮ | ⋮ | ⋮ | ⋮ | ⋱ |

**Table 10.** Results recovered by Algorithm 2.

| Target Label | Platf-1 FW | PW | PT | AM | AOA | Platf-2 FW | PW | PT | AM | DOA | · · · |
|---|---|---|---|---|---|---|---|---|---|---|---|
| 1 | $4.7654 \times 10^3$ | 2.1200 | 826.1695 | 95.2850 | 0 | $4.8275 \times 10^3$ | 1.7705 | 802.9382 | 95.7800 | 428.7967 | · · · |
| 2 | $4.7653 \times 10^3$ | 1.8760 | 800 | 95.5050 | 480.4501 | $4.7657 \times 10^3$ | 1.8000 | 800.3337 | 95.5050 | 428.7933 | · · · |
| 3 | $4.7655 \times 10^3$ | 1.8730 | 822.1822 | 95.5600 | 428.7933 | $4.7656 \times 10^3$ | 1.9000 | NaN | 95.5050 | 428.7767 | · · · |
| 4 | $4.7656 \times 10^3$ | 1.8000 | 830 | 93.6472 | 428.7867 | $4.7655 \times 10^3$ | 1.9200 | 760 | 96.0750 | 428.8000 | · · · |
| 5 | $4.7656 \times 10^3$ | 1.8156 | 796.9825 | 95.5600 | 464.9850 | $4.7655 \times 10^3$ | 2.1200 | 780 | 95.5050 | 428.7633 | · · · |
| 6 | $4.7653 \times 10^3$ | 2.0800 | 830 | 93.6649 | 428.7700 | $4.7656 \times 10^3$ | 1.7131 | 776.9018 | 95.6700 | 428.7867 | · · · |
| 7 | $4.7654 \times 10^3$ | 1.8653 | 818.7611 | 95.6700 | 477.6912 | $4.7842 \times 10^3$ | 1.7400 | 770 | 95.8159 | 470.2310 | · · · |
| 8 | $4.7655 \times 10^3$ | 2.2000 | 840 | 95.6700 | 428.7833 | $4.7656 \times 10^3$ | 1.7600 | 783.2492 | 95.6700 | 462.8508 | · · · |
| 9 | $4.7506 \times 10^3$ | 2.2000 | 813.6863 | 95.3412 | 428.8000 | $4.7656 \times 10^3$ | 1.9000 | 790 | 95.2221 | 428.7733 | · · · |
| 10 | $4.7656 \times 10^3$ | 1.9000 | 840 | 95.6150 | 468.3993 | $4.7656 \times 10^3$ | 1.9000 | 800 | 95.6150 | 428.8000 | · · · |
| 11 | $4.7656 \times 10^3$ | 1.9200 | 807.3440 | 94.5981 | 428.8000 | $4.7656 \times 10^3$ | 1.9000 | 784.6421 | 95.5050 | 428.8000 | · · · |
| 12 | $4.7656 \times 10^3$ | 1.8600 | 870 | 95.6633 | 428.8000 | $4.7655 \times 10^3$ | 1.7497 | 800 | 95.5600 | 428.8267 | · · · |
| 13 | $4.7653 \times 10^3$ | 1.8397 | 807.5579 | 95.5600 | 428.7700 | $4.7657 \times 10^3$ | 1.7306 | 784.8500 | 94.5049 | 428.7733 | · · · |
| 14 | $4.7536 \times 10^3$ | 2.1200 | 880 | 95.4003 | 428.7833 | $4.7656 \times 10^3$ | 1.8800 | 810 | 95.5050 | 428.8000 | · · · |
| 15 | $4.7655 \times 10^3$ | 2.1400 | 811.4794 | 95.5600 | 428.8000 | $4.7656 \times 10^3$ | 1.7390 | 810 | 95.6700 | 428.7433 | · · · |
| 16 | $4.7656 \times 10^3$ | 2.1600 | 890 | 95.6150 | 428.7900 | $4.7656 \times 10^3$ | 1.8000 | 794.3686 | 95.6510 | 428.7933 | · · · |
| 17 | $4.7656 \times 10^3$ | 2.1600 | 900 | 95.5600 | 428.7933 | $4.7656 \times 10^3$ | 1.7596 | 820 | 95.4500 | 428.7767 | · · · |
| 18 | $4.7656 \times 10^3$ | 1.9200 | 890 | 95.7352 | 428.7767 | $4.7742 \times 10^3$ | 1.8800 | 830 | 95.5600 | 428.8000 | · · · |
| 19 | $4.7610 \times 10^3$ | 1.8809 | 730 | 96.7416 | 481.7037 | $4.7656 \times 10^3$ | 1.7994 | 830 | 95.5600 | 428.8000 | · · · |
| 20 | $4.7654 \times 10^3$ | 1.9000 | 826.6687 | 95.5050 | 482.3048 | $4.7655 \times 10^3$ | 2.1400 | 830 | 95.6150 | 428.7533 | · · · |
| ⋮ | ⋮ | ⋮ | ⋮ | ⋮ | ⋮ | ⋮ | ⋮ | ⋮ | ⋮ | ⋮ | ⋱ |

The correction for wrong PDW records of Algorithms 1 and 2 are also validated. For the real data $X$ in Table 7, it can be seen that the PW and AOA records of Target "19" for platform "1" are "0.2200" and "0.3533" with underline, which is wrong and totally different from other platform records. From Table 9, we have that the correction of Algorithm 1 for PW and AOA are "1.7863" and "461.7466", which are close to the records of platform "2". The results of Algorithm 2 are consistent with Algorithm 1, which suggests that the proposed methods can correct the wrong records efficiently.

In addition, the run times of Algorithms 1 and 2 are compared under different missing ratios, and the result is shown in Figure 4. We see that the run time of Algorithm 2 is stable for different missing ratios, and much lower than Algorithm 1 when the missing ratio exceeds 0.3. This is consistent with the complexity analysis at the end of Section 4.3.

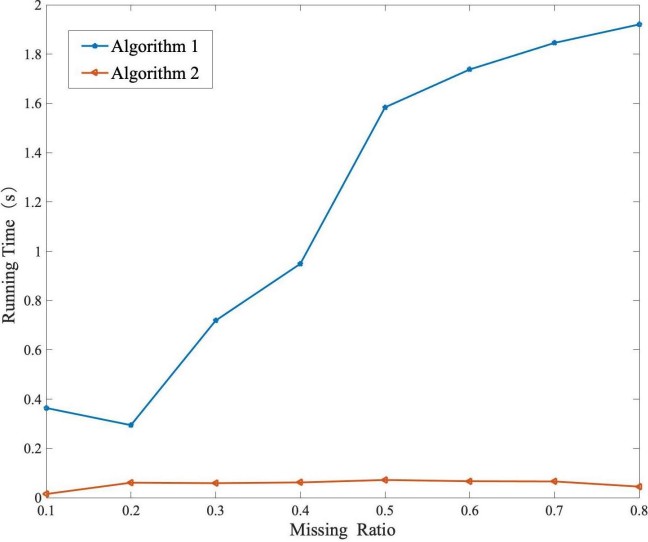

**Figure 4.** The running time comparison of Algorithms 1 and 2 under different missing ratios.

In the end, the iteration number of Algorithms 1 and 2 under different missing ratios are shown in Figure 5, it can be found that the iteration number of Algorithm 2 is lower than Algorithm 1 and stable in different missing ratios, which is consistent with the running time and complexity analysis.

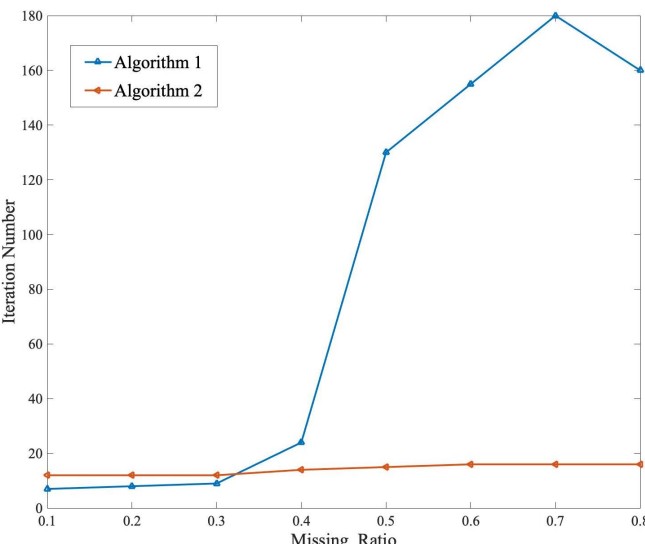

**Figure 5.** The iteration number comparison of Algorithms 1 and 2 under different missing ratios.

*5.3. Comparison Test*

In this section, the comparison test of proposed methods with three state-of-the-art methods for electromagnetic data annotation completion is given. Three compared methods are:

1. The K-nearest neighbor method (KNN) in [32], which predicts the missing annotation by its K nearest neighbors;
2. The augmented Lagrange multiplier method for low-rank matrix recovery (ALM) in [27], where the annotation completion is formulated as a convex optimization model solved by the ALM algorithm;
3. The nuclear norm regularized method for annotation completion (NNLS) in [28], where the annotation completion is formulated as an optimization model solved by the accelerated proximal gradient algorithm.

For comparison testing, the synthetic data is utilized, which is generated by the radar target simulator with 10 platforms, 10 targets, and in $t = [t_1, \ldots, t_{10}]$, 10 features are utilized for each target, which forms the original data matrix $X$ with size $100 \times 100$ and rank $r = 10$.

Then, the performance of the proposed methods and compared methods are discussed. The MSE of five methods under different missing ratios are shown in Figure 6. It can be seen that the MSE increases roughly with the increase of the missing ratio for all methods. The MSE of proposed Algorithms 1 and 2 are roughly the same, and much lower than the KNN, ALM, and NNLS methods, which demonstrates the superior recovery performance by using the ADMM algorithms. Compared to the KNN with Algorithms 1 and 2, it can be found that utilizing the low-rank structure for annotation completion can recover the missing annotation efficiently. In addition, the average MSE of the five compared methods is presented in Table 11. For each missing ratio, the feature parameters are dropped randomly ten times to get the average MSE of different compared methods.

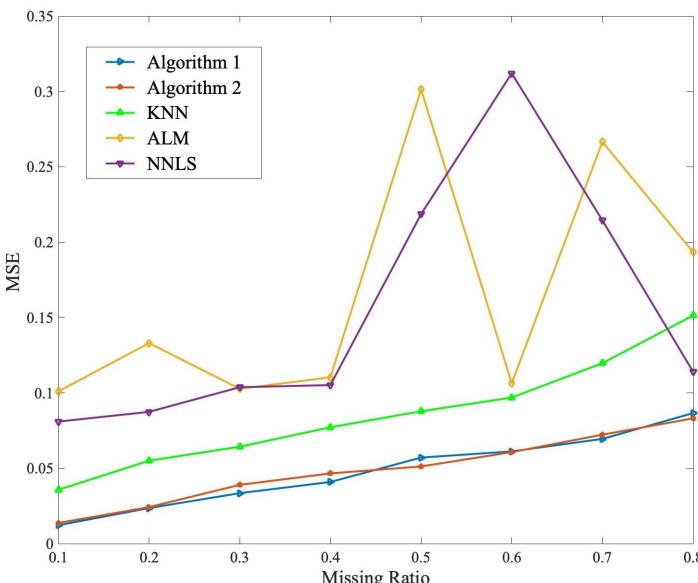

**Figure 6.** The MSE comparison for different methods under different missing ratios.

In the end, the running time for different methods is given in Figure 7. We find that the proposed Algorithm 1 is more time-consuming than other compared methods since the SVD decomposition, and the running time is much more with the increasing of missing ratio. The KNN method has the lowest running time since the low computation. The running time of NNLS and ALM methods are lower than the proposed method's Algorithms 1 and 2; the main reason is that the SVD decomposition in the proposed algorithms is time-consuming.

**Table 11.** The average MSE of five compared methods.

| Missing Ratio | Algorithm 1 | Algorithm 2 | KNN | ALM | NNLS |
| --- | --- | --- | --- | --- | --- |
| 0.1 | 0.0122 | 0.0137 | 0.0357 | 0.0810 | 0.1010 |
| 0.2 | 0.0236 | 0.0242 | 0.0550 | 0.0874 | 0.1330 |
| 0.3 | 0.0355 | 0.0390 | 0.0643 | 0.1038 | 0.1027 |
| 0.4 | 0.0409 | 0.0466 | 0.0772 | 0.1052 | 0.1104 |
| 0.5 | 0.0571 | 0.0512 | 0.0878 | 0.2186 | 0.3014 |
| 0.6 | 0.0607 | 0.0612 | 0.0969 | 0.3120 | 0.1064 |
| 0.7 | 0.0695 | 0.0723 | 0.1197 | 0.2144 | 0.2665 |
| 0.8 | 0.0866 | 0.0831 | 0.1513 | 0.1141 | 0.1934 |

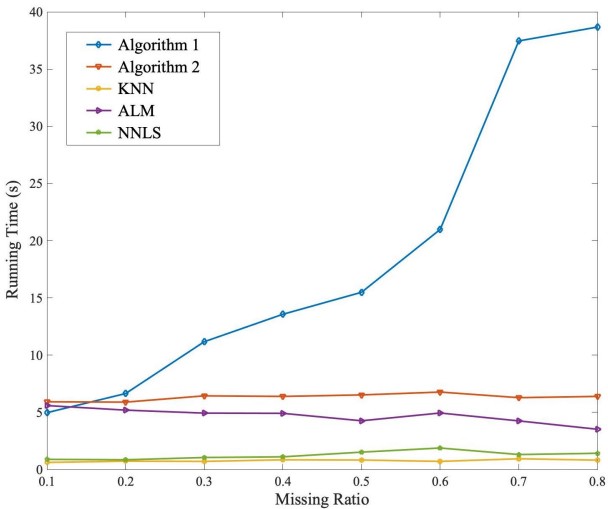

**Figure 7.** The running time comparison for different methods under different missing ratios.

Based on the discussion above, we have that the proposed methods can recover and correct the missing and wrong annotation efficiently, but the running time is much more than compared methods.

## 6. Conclusions

In this work, we have considered cooperative annotation for electromagnetic reconnaissance data. By exploiting the correlation of observations at different platforms, we formulate the annotation completion problem as a low-rank matrix recovery problem and proposed two methods to solve this problem, including the rank-minimization-based convex algorithm and the maximum-rank-decomposition non-convex algorithm. Numerical experiments on synthetic data and real data suggest that the proposed methods can recover the missing annotation efficiently and achieve better MSE performance than the compared annotation methods.

**Author Contributions:** Conceptualization, W.Z. and J.Y.; methodology, W.Z., Q.L. and G.S.; software, Q.L. and J.L.; validation, W.Z., H.S. and G.S.; visualization, Q.L. and H.S.; writing—original draft, W.Z. and G.S.; writing—review and editing, J.Y. and Q.L. All authors have read and agreed to the published version of the manuscript.

**Funding:** This research was funded by the National Natural Science Foundation of China (NSFC) 61871092 and U20B2070.

**Data Availability Statement:** Not applicable.

**Conflicts of Interest:** The authors declare no conflict of interest.

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
