# Peer review of "Cooperative Electromagnetic Data Annotation via Low-Rank Matrix Completion"

_remotesensing, doi:10.3390/rs15010121_

Round 1

Reviewer 1 Report

On page 5/18 between formulae (11) and (12) LaTeX did not take a reference.
The mathematical formulation is clear. The indication of the dimensionality of the parameters and its domain of existence is missing.
I ask you a question. Does Latecnica also work for complex data? Report the answer in the article.
Increase the bibliography concerning the use of L+S in compound low-rank matrices.
Also for mathematics, sometimes it is not clear whether the parameters are matrices or vectors. Please use symbols to differentiate their nature.
Why are there NaNs in Table 7? Please explain further.
Everything else is OK.

Reviewer 2 Report

1. This paper proposes two methods, including the rank-Minimization-Based convex approximation algorithm and the maximum-Rank-Decomposition-Based Non-Convex algorithm, and both of them have been evaluated. Can these two methods be integrated into one method? 2. It is introduced in the abstract and summary of the paper that the proposed methods can not only recover missing data, but also correct wrong data, but there is no related content in the paper? 3. Figure 3 is only the result of one case in Table 5. It is suggested to give the statistical results for different deletion rates. 4. In the section of method evaluation based on simulation and measured data, the evaluation of the method will be more convincing if the statistical results of various situations are given. 5. Does the table on page 8 and the following pages take up too much space? Whether there is a more appropriate expression. In addition, section 3.1, above Formula (12) "Then, the optimal solution of problem (8) is given by [?] ",there is an extra "[?] ".

Reviewer 3 Report

Authors -- This is a very well written paper.  This was very easy to review.  You took a lot of time making sure the presentation was smooth and without errors.  Your hard work is to be applauded.  There are a minor number of word choice differences I would like you to consider; listed below:

 Line 51 -- "...which brings obstacle..." change to "...which presents and obstacle..."

 Line 67 -- "...recovery of low-rank matrix..." change to "...recovery of the low-rank matrix..."  -- add the word "the"

 Line 70 -- "...completion [22-30], but to the..." change to "...completion [22-30], to the..."  -- remove the word "but"

 Line between equations (11) & (12) -- "...optimal solution of problem (8) is given by [?]..."  What does the symbol "?" refer to ?

 Line between equations (13) & (14) -- "... which should be set zeros to..."  change to "...which should be a set of zeros to ..."

 Line 181 -- add ":" after word "are"; should read "...methods are:..."

 Line 202 -- change "...Algorithm 2..." notation to "...Algorithm 1..."; I think that is what you meant but I'm not sure.

There are no more comments.  Well Done !!

Round 2

Reviewer 1 Report

Accepted
